# Risk factors for vitamin D deficiency in Abu Dhabi Emirati population

**Amal Abdul Rahim Al Zarooni**[1]*, **Nico Nagelkerke**[2], **Fatima Ibrahim Al Marzouqi**[3], **Salma Hamad Al Darmaki**[4]

**1** Ambulatory Healthcare Services, Abu Dhabi Health Services Company, Al Ain, United Arab Emirates, **2** Community Medicine, College of Medicine and Health Sciences, United Arab Emirates University, Al Ain, United Arab Emirates, **3** Ministry of Health and Prevention, Ajman, United Arab Emirates, **4** Tawam Hospital, Seha, Al Ain, United Arab Emirates

* a.alzarouni2019@gmail.com

## Abstract

### Objective

There have been concerns regarding the high prevalence of vitamin D deficiency in the United Arab Emirates and the association between this prevalence and cardiovascular risk. Vitamin D deficiency is a global public health problem, with a high prevalence in the Abu Dhabi national population. This cross-sectional retrospective observational study aimed to elucidate vitamin D deficiency symptoms and contributing factors in this context.

### Method and sampling

Among 12346 participants presenting for their Weqaya population comprehensive cardiovascular risk factor screening, 700 were randomly selected and telephoned, among whom 400 completed a questionnaire that elicited information on dietary intake, sun exposure, clothing, physical activity, and symptoms related to vitamin D deficiency.

### Results

Higher vitamin D levels were noted in participants who consumed extra servings of cod liver oil, tuna, salmon, and eggs; who wore clothes with their arms and legs exposed; and who performed daily physical activity. Symptoms of hypovitaminosis D (bone pain) were more frequently found in females. Fatigue is significantly associated with low vitamin D and younger participants. Strategies focusing on health promotion and lifestyle interventions should be implemented to address vitamin D deficiency.

## Introduction

Over the past decades, vitamin D deficiency has increased rapidly worldwide in children and adults [1–3]. About 30%–50% of children and adults in the United Arab Emirates (UAE), Australia, Turkey, India, and Lebanon have 25-hydroxyvitamin D levels <20 ng/ml [1, 4, 5].

**Data Availability Statement:** All relevant data are within the manuscript and its Supporting Information files.

**Funding:** The author(s) received no specific funding for this work.

**Competing interests:** The Seha company is the government institution and had no role in data collection or analysis or writing the manuscript and no conflict of interest The authors have declared that no competing interests exist.

**Abbreviations:** UAE, United Arab Emirates.

The increase in the prevalence of vitamin D deficiency may be due to low skin exposure to ultraviolet radiation [1, 6], and this is a significant problem in Arab countries where women dress conservatively, covering most of their bodies when outdoors [1, 7, 8]. In a recent study, Arab-American women who observed a conservative dress style had a significantly higher prevalence of vitamin D deficiency than their counterparts who dressed less conservatively [9, 10]. In UAE, young women similarly observe a very conservative style of dressing that restricts sunlight exposure [9].

Low dietary intake of vitamin D and calcium, and other factors, including obesity and low social status, are all associated with low serum levels of vitamin D [1, 11, 12].

Daytime outdoor physical activity may serve as a surrogate indicator for sun exposure. It is also well established that physical exercise increases local bone mass, reduces calcium excretion, and increases absorption efficiency, thus increasing serum calcium levels, which results in sparing serum vitamin D. Additionally doing physical activity will lead to increases in the lipolysis rate, which stimulates the mobilization of vitamin D from adipose tissues, thereby raising its serum level [1, 13, 14]. It has been reported that, at least among adults, women have higher rates of induced systemic lipolysis than men [1, 15].

There was a significant association between 25OHD levels and total hip BMD in both younger and older adults participating in the NHANES III survey as recently reported [16]. Additionally, supplementation with vitamin D can improve BMD and reduce fractures [2, 17, 18].

This study aimed to assess the Association of vitamin D levels with age, sex, education level, dietary intake, duration of sun exposure, type of clothing, physical activity, body ache symptoms, history of fractures, family history of osteoporosis, adherence, and compliance to vitamin D treatments.

## Materials and methods

### Study design

Cross-sectional retrospective observational study conducted in Ambulatory health care services, Abu Dhabi, United Arab Emirates.

### Subject

Adults aged ≥18 years presenting to Ambulatory Health Care clinic for the Weqaya screening program between October 2011 and February 2013 were randomly recruited [19]. The Inclusion Criteria are participants who did the Weqaya screening test and agreed to be enrolled in the study after taking verbal consent through the phone. The only exclusion criteria were refusing to participate, impossibility to reach by phone, and having chronic diarrhea, chronic liver disease, and chronic kidney disease, i.e. conditions that may severely affect Vitamin D levels. The study protocol was approved by the Human Research Ethics Committee of Al Ain Medical District. Written informed consent was obtained at the time of the Weqaya screening test. Follow up appointment for result discussion, counseling, and treatment was done by health care providers in primary health care centers.

Weqaya screening program is an Abu Dhabi government screening program that identifies cardiovascular risk factors and improves screeners' health status. All adults (aged >18 years) UAE nationals living in Abu Dhabi Emirate are eligible for this screening every three years. In addition, body mass index, blood pressure, smoking status, glycated hemoglobin, total cholesterol/high-density lipoprotein cholesterol ratio, vitamin D level, and creatinine level were measured as part of the program.

## Data collection

Data were collected from the Weqaya register, and questionnaires were completed via phone interviews.

## Sampling and methods

Among 12346 participants who underwent Weqaya screening between October 2011 and November 2012, 700 were randomly selected using a computer-generated sequence and were telephoned. Among the 700 participants, 300 were excluded because they either did not answer their phone (160), had their phones switched off (120), or refused to participate (20), and 400 participants agreed to complete the questionnaires. As a final step, eight participants with liver diseases were excluded from the remaining 400 participants. Data of the final 392 participants were analyzed. The data are available as S1 File.

## Questionnaire

Eligible participants were interviewed via phone to complete their questionnaires. The questioner was designed based on previous studies and literature reviews. The survey was in English language and translated to the Arabic language by two researchers independently. The researcher explained to the participant the aim and objective of the study, and they took verbal informed consent with providing information about the confidentiality and anonymity of the Participants. Questionnaires included information about basic demographics, education level, and lifestyle factors (e.g., sunlight time exposure, physical activity, sleeveless clothing, sunblock, and dietary intake of dairy products and food rich in vitamin D). In addition, questions about taking multivitamin, vitamin D supplements, and calcium were included.

Vitamin D deficiency symptoms (e.g., bone pain, back pain, muscle pain, and fatigue) were also covered. Personal and family history of fractures, osteoporosis, adherence to treatment, and any medication side effects were also included. Additionally, a history of chronic medical diseases, including liver disease, kidney disease, and chronic diarrhea were assessed.

The questionnaires were validated in a pilot study comprising ten patients from ambulatory health centers who were not included in the current study. The questions were then accordingly adjusted. The questionnaires are available as S1 and S2 Questionnaires.

## Statistical analysis

We calculated the sample size on the basis of 80% power and a 5% two-tailed significance level. We assumed that we would need to explore modest differences of a continuous variable between two groups of unequal size. We therefore calculated the sample size on the basis of detecting a difference of half a standard deviation between two groups the smallest of which comprises 10 of the totals. This yielded a required sample size of 350 (https://epitools.ausvet.com.au/twomeanstwo). We then doubled this to take into account factors such as refusals and non-response.

Data were analyzed using the SPSS program version 19 (IBM Corp. Released 2010. IBM SPSS Statistics for Windows, Version 19.0. Armonk, NY: IBM Corp). Besides standard descriptive and basic analytical statistics such as Chi-square tests, two multivariate statistical methods, linear regression to assess the determinants of vitamin D level and binary logistic regression to identify determinants of binary outcomes, were used. A significance level of 0.05 was used throughout.

## Results

The data of the 392 eligible participants were analyzed. Women accounted for 61.5% of participants. Participants' ages ranged between 18 and 70 (mean, 38.9, SD 13.2) years. Regarding education level, most participants had school education only (52.5%). School education reported were primary school (10.2%), intermediate school (13%) and Secondary School (29.3%). Other education levels reported were Ph.D./Masters (2.3%), College/University (28.6%), and informal education (i.e., no formal schooling) (14.4%). Approximately 94.1% of the participants were from urban areas. The following cutoff values for vitamin D were defined: deficient ($<$ 50 nmol/L), insufficient (50–74 nmol/L) and normal ($\geq$ 75 nmol/L). Vitamin D levels varied among participants, and most participants (79.6%) had deficient vitamin D levels. Only 23% of participants consumed $>$3 servings of dairy products per week, and 19.4% consumed $>$3 servings of cod liver oil or tuna per week. Only one-fourth of participants reported sun exposure. More than 50% of participants never wore sleeveless clothing. Regarding physical activity, 26.8% of participants performed some type of physical activity daily, 7.9% weekly, and 22.8% never exercised. Approximately 53.6% of participants were unemployed (housewives, retired, and students). Among the participants, 40% had chronic medical illnesses, mostly diabetes mellitus, dyslipidemia, and hypertension (**Table 1**).

### Age, gender, education level, and vitamin D levels

Serum vitamin D levels were regressed linearly on age, gender, and education level (coded 1–6 from low to high) revealed that vitamin D levels were significantly higher in older participants (Beta = 0.497 nmol/L/year, P $<$ .001) and significantly lower those with a higher education level (Beta = -2.032; P = .017). Vitamin D levels were also lower in female participants, although this finding was not statistically significant.

### Dietary intake and vitamin D levels

Regarding dietary intake, linear regression showed that participants who consumed extra servings of cod liver oil, tuna, salmon, and egg had higher vitamin D levels (P = 0.014). For each extra serving of these dietary components, vitamin D levels increased by 2.4 nmol/L. Moreover, the intake of dairy products led to an increase in vitamin D levels, although insignificantly.

### Sun exposure, clothing, physical activity, and vitamin D levels

By linear regression, after adjustment for age, participants exposed to the sun between 9 am and 5 pm had slightly higher vitamin D levels, but this did not reach statistical significance.

Participants who used sunscreen products had lower vitamin D levels, but this association did not reach statistical significance. Regarding clothing habits, participants who wore clothes with their arms and legs exposed had higher vitamin D levels (P = .005).

We then carried out multiple linear regression of vitamin D levels on the most important covariates identified above, age, education level, food servings of tuna, cod liver oil, salmon and egg, and clothing that allows sun exposure (e.g., sleeveless clothes). (Table 2)

According to the physical activity, participants with daily physical activity had the highest mean vitamin D levels (38.2 nmol/L). In contrast, the lowest mean vitamin D level was noted in participants who rarely do physical activity (31.7 nmol/L).

Depending on the nature of their work, participants who spent most of their time sitting while at work had the lowest mean vitamin D levels (29.6 nmol/L). In contrast, the highest mean vitamin D levels were noted in participants whose work necessitated physical effort (e.g., handling heavy objects) (37.1 nmol/L) with a significant P value of 0.045.

**Table 1. Sociodemographic characteristics.**

| | Character | Percentage |
|---|---|---|
| Vitamin D level | Normal ($\geq$ 75 nmol/L) | 5.1% |
| | Insufficiency (50–74 nmol/L | 15.3% |
| | Deficiency (< 50 nmol/L) | 79.6% |
| Gender | Female | 61.5% |
| | Males | 38% |
| Education level | PHD/Master | 2.3% |
| | College/University | 28.6% |
| | School | 52.5% |
| | primary school (10.2%) | |
| | intermediate school (13%) | |
| | Secondary School (29.3%) | |
| | No formal education | 14.4% |
| City/facility | Urban | 94.1% |
| | Suburban | 5.9% |
| Dairy product servings | >3/week | 23% |
| | $\leq$3/week | 76.6% |
| Cod liver oil/tuna servings | >3/week | 19.4% |
| | $\leq$3/week | 80.4% |
| Sun exposure | Yes | 24.2% |
| | No | 75.3% |
| Sleeveless clothing | Never | 53.3% |
| | Rarely | 20.4% |
| | Sometimes | 18.1% |
| | Most of the time | 7.9% |
| | Always | 0.3% |
| Physical activity | Daily | 26.8% |
| | >1/week | 26.3% |
| | Weekly | 7.9% |
| | Rarely | 16.6% |
| | Never | 22.8% |

## Symptoms of vitamin D deficiency

Bone pain as a symptom of hypovitaminosis was noted more frequently in female participants (126/242 vs. 48/150) with a significant P value of <0.001. Muscle ache was statistically more common in females (112/242 vs. 57/150), but differences did not reach statistical significance. Similarly, back pain was found more often in females (110/242 vs. 59/150); however, these differences did not reach statistical significance. Fatigue was found more frequently in female participants (105/241 vs. 48/150), but this did not reach statistical significance. There was no

**Table 2. Linear regression for association of vitamin D levels with age, education level, food serving, and clothing habit.**

| | B | t | P-value |
|---|---|---|---|
| Age (years) | .288 | 5.136 | p<0.0001 |
| Education level (1 = No formal education, 6 = PHD/Master) | -.140 | -2.490 | 0.013 |
| Food serving (tuna, cod liver oil, salmon, and egg) | .126 | 2.732 | 0.007 |
| Clothing (sleeveless shirt/short) | .098 | 2.109 | 0.036 |

**Table 3. Logistic regression for association of osteoporosis with age, gender, and vitamin D level.**

|  | B | S.E. | Wald | df | P-value | Exp(B) |
|---|---|---|---|---|---|---|
| Age (years) | .156 | .026 | 35.167 | 1 | 0.000 | 1.169 |
| Gender | -2.463 | .662 | 13.855 | 1 | 0.000 | .085 |
| Vitamin D Level | .005 | .009 | .287 | 1 | 0.592 | 1.005 |

correlation between the mean vitamin D level and vitamin D deficiency symptoms (e.g., bone pain, muscle ache, and back pain), and only fatigue was found to be significantly associated with low vitamin D levels (P-value 0.0480). There was no correlation between the age and symptoms of vitamin D deficiency (e.g., bone pain, muscle ache, and back pain), and only fatigue was found to be significantly associated with younger participants (P-value 0.010).

### Fractures and vitamin D levels

We found that fractures were reported significantly more among male participants (29/150) than among female participants (24/242) (P value 0.012). This result was possible because male individuals are more involved in road traffic accidents and sports injuries. We also used logistic regression to regress a history of fractures on gender, age, and vitamin D levels, but only gender turned out to be a significant predictor of fractures.

### Osteoporosis and vitamin D levels

By logistic regression, osteoporosis increased significantly with age and was significantly more prevalent among females than males but was not significantly associated with Vitamin D levels. (Table 3)

Participants who had a maternal history of hip fracture had lower vitamin D levels (P-value 0.04).

### Complains to vitamin D treatment

A total of 62.7% of participants received vitamin D treatment, and 85% were compliant. However, of those who took the treatment, 9.2% had missed at least one tablet throughout their treatment course. Compliance problem was more common with male gender, younger age, lower vitamin D level and higher education level with insignificant P-value.

Of all the participants who took vitamin D treatment, only 4.8% reported side effects mainly gastroenterology-related, e.g., abdominal pain, abdominal distention, diarrhea, and flank pain.

### Discussion

This study informed us of vitamin D deficiency determinants in Abu Dhabi citizens. It included sun exposure, degree of physical activity, clothing, and dietary habits. Younger participants also had a significantly higher risk of prevalence of vitamin D deficiency, which is worth studying as factors such as changes in housing style from traditional houses to modern flats and villas, may lead to less sun exposure. In addition, dietary factors may contribute as the current generation consumes more fast food, lacking essential vitamins and minerals.

Other lifestyle factors may play a role, such as the new digital technologies, which may affect the younger generation by decreasing their outdoor activities and increasing time spent indoors using these technologies (e.g., electronic games, smartphone devices, computers, and televisions). Moreover, type of occupation; office employment compared to less building bound occupation. The finding supports that education level and employment status were a

major determinant of vitamin D levels, possibly because highly educated participants practice indoor office-based work with less sun exposure. Health-seeking behavior may contribute, especially that compliance with medications was lower in the younger age group. This was found in other countries as urbanization and practicing indoor lifestyle in developed countries contribute to hypovitaminosis [20]. Furthermore, highly educated participants have problems with medication compliance, as observed in our results. Participants involved in heavy out-door jobs have higher vitamin D levels because they are more exposed to sunlight.

There may be seasonal factors worth future study. As 75% of participants did not expose themselves to sunlight, especially during the summer when daytime temperatures usually exceed 40˚C. This limits the time spent outdoors during the day. A study conducted in UAE for seasonal variation show that the mean vitamin D levels were highest in the winter. However, the mean vitamin D levels were at its lowest in the hottest months in the summer [21].

The duration of sunlight exposure necessary to maintain adequate stores of vitamin D has always been unclear. However, the duration was recently recommended to be 10–15 min of mid-day sun (10:00 am–3:00 pm) [16]. Our study found no significant association between daytime (9 am– 5 pm) sunshine exposure and vitamin D levels. Most likely because people rarely expose their skin while out in the sun for cultural and religious reasons. Deliberate sun exposure was not assessed as preventive or therapeutic action. A different study design is needed for confirmation of this area [22].

The lack of an association between Vitamin D levels and osteoporosis and fractures may be due to confounding by indication: finding could be that osteoporotic (fracture) participants had sought more medical advice and became aware of bone strengthen factors. In addition, they could be on calcium and vitamin D supplementations. Further studies are needed to investigate these findings. A study conducted in elderly Chinese patients shows that vitamin D insufficiency and deficiency was higher in patients with osteoporosis than in those without osteoporosis and suggests that vitamin D plays an essential role in hip fractures in elderly patients [23].

Different generations as older are more active in their youth and were exposed to the sun more at the stage of mineralization. Therefore, we may see different outcomes in the younger population with different dietary and lifestyle habits.

## Conclusion

Vitamin D deficiency is a global public health problem with a high prevalence in the Abu Dhabi population. Therefore, strategies focusing on health promotion and lifestyle intervention should be implemented to address the major problems of vitamin D deficiency.

### Recommendations

Regarding the looming epidemic of vitamin D deficiency, the following strategies are recommended:

- Promotion of outdoor lifestyle activities within communities

- Establishment of private female compounds that facilitate sun exposure

- Implementation of a school project that promotes an active lifestyle, sun exposure, and intake of vitamin D-rich diet

- Establishment of government policies for food fortification with vitamin D

- Encouragement of health education and prevention of vitamin D deficiency

- Involvement of media in increasing awareness about the alarming prevalence of vitamin D deficiency in the UAE society and its serious consequences

## Limitations

One of the strong points of the current study was that our study sample represented the Abu Dhabi national population. Our study had few limitations. It was a retrospective study; thus, the data based on the questionnaires were subject to recall bias. The study did not assess parathyroid hormone and bone profile (calcium, phosphate, and alkaline phosphates) to exclude secondary hyperparathyroidism, although other secondary causes of vitamin D deficiency (e.g., chronic kidney disease, liver disease, and malabsorption) were excluded. The electronic-generated report for vitamin D treatment was limited to government clinics. Private institutions were not included. The UAE population is a mixture of different nationalities and ethnic races, and our study focused on UAE nationals only, mainly with light skins.

Further studies are needed to compare UAE nationals with expatriates. One of the difficulties that we faced while conduction this study was that 280 participants among 700 were not reachable via the phone (either their phones were switched off or they did not answer), although two attempts were made to contact them. Only if participation was associated with vitamin D levels and the determinants that we explored could this have biased associations.

## Supporting information

**S1 File.**
(SAV)

**S1 Questionnaire.**
(DOC)

**S2 Questionnaire.**
(DOCX)

## Acknowledgments

The authors appreciated the assistance and support of
  Dr. Latifa Al Ketbi
  Mr. Khalil AL Rahman
  Dr. Shamma Alawai
  Ms. Amna Saeed
  Ms. Anoud Al Shamsi
  Ms. Moza ALDhaheri
  Ms. Sana Zein AlDin

## Author Contributions

**Conceptualization:** Amal Abdul Rahim Al Zarooni, Fatima Ibrahim Al Marzouqi, Salma Hamad Al Darmaki.

**Data curation:** Amal Abdul Rahim Al Zarooni, Fatima Ibrahim Al Marzouqi, Salma Hamad Al Darmaki.

**Formal analysis:** Amal Abdul Rahim Al Zarooni, Nico Nagelkerke, Fatima Ibrahim Al Marzouqi, Salma Hamad Al Darmaki.

**Methodology:** Amal Abdul Rahim Al Zarooni.

**Supervision:** Nico Nagelkerke.

**Writing – original draft:** Amal Abdul Rahim Al Zarooni, Fatima Ibrahim Al Marzouqi, Salma Hamad Al Darmaki.

**Writing – review & editing:** Amal Abdul Rahim Al Zarooni, Nico Nagelkerke, Fatima Ibrahim Al Marzouqi, Salma Hamad Al Darmaki.

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
