## [Decision Letter · Decision Letter 0]

20 Oct 2021

PONE-D-21-12504Vitamin D deficiency-associated risk factors and symptoms in the Abu Dhabi Emirati populationPLOS ONE

Dear Dr. Amala Al Zarooni,

Thank you for submitting your manuscript to PLOS ONE. After careful consideration, we feel that it has merit but does not fully meet PLOS ONE’s publication criteria as it currently stands. Therefore, we invite you to submit a revised version of the manuscript that addresses the points raised during the review process. Please submit your revised manuscript by October 27, 2021. If you will need more time than this to complete your revisions, please reply to this message or contact the journal office at plosone@plos.org. Please include the following items when submitting your revised manuscript:A rebuttal letter that responds to each point raised by the academic editor and reviewer(s). You should upload this letter as a separate file labeled 'Response to Reviewers'.A marked-up copy of your manuscript that highlights changes made to the original version. You should upload this as a separate file labeled 'Revised Manuscript with Track Changes'.An unmarked version of your revised paper without tracked changes. You should upload this as a separate file labeled 'Manuscript'.

We look forward to receiving your revised manuscript.

Kind regards,

Marcello Ciaccio, M.D., Ph.D

Academic Editor

PLOS ONE

Journal Requirements:

2. In your Methods section, please provide a justification for the sample size used in your study, including any relevant power calculations (if applicable).

Please include additional information regarding the survey or questionnaire used in the study and ensure that you have provided sufficient details that others could replicate the analyses. For instance, if you developed a questionnaire as part of this study and it is not under a copyright more restrictive than CC-BY, please include a copy, in both the original language and English, as Supporting Information. In addition please provide additional information regarding the interview guide development process, including the theories or frameworks which were employed.

We note that you have reported significance probabilities of 0 in places. Since p=0 is not strictly possible, please correct this to a more appropriate limit, eg 'p<0.0001'.

4. Thank you for stating the following in the Financial Disclosure section: 

We note that one or more of the authors are employed by a commercial company: Ambulatory Healthcare Services, Abu Dhabi Health services company. 

Reviewers' comments:

Reviewer's Responses to Questions

**Comments to the Author**

1. Is the manuscript technically sound, and do the data support the conclusions?

Reviewer #1: Yes

Reviewer #2: Yes

2. Has the statistical analysis been performed appropriately and rigorously? 

Reviewer #1: I Don't Know

Reviewer #2: Yes

3. Have the authors made all data underlying the findings in their manuscript fully available?

Reviewer #1: Yes

Reviewer #2: Yes

4. Is the manuscript presented in an intelligible fashion and written in standard English?

Reviewer #1: Yes

Reviewer #2: Yes

5. Review Comments to the Author

Reviewer #1: The title must be changed because it is not clear

Material and Methods:

I suggest to begin by stating the study design and where the study was performed. Please, specify all inclusion and exclusion criteria.

Please, specify were the study was conducted.

Please, specify the instrumentation for immunoturbidimetry assay.

Please, verify that the following sentence is correct “Cardiac C-Reactive Protein (Latex) High Sensitive kit (CRPHS)”.

In the material and methods, Authors should state that there was also a control cohort (non-HTN) specifying the characteristic of such population, including the criteria of exclusion and inclusion

The discussion should be better organized and also more appropriate terms should be used.

Reviewer #2: The manuscript cover a large part of Vitamin D deficiency issus, addressing many questions in an elegant fashion and a clear presentation. However, some minor changes could improve it: please, revise the reference list according to the major works thatare available in the literature, mainly consider citing: -PMID: 18516267 -PMID: 31330127 -PMID: 28801380 -PMID: 30588177 -PMID: 31284484 -PMID: 30793054 -PMID: 31142227

6. PLOS authors have the option to publish the peer review history of their article (what does this mean?). If published, this will include your full peer review and any attached files.

Reviewer #1: No

Reviewer #2: No

---

## [Author Response · Author response to Decision Letter 0]

28 Oct 2021

requirements.

Done

2. In your Methods section, please provide a justification for the sample size used in your study, including any relevant power calculations (if applicable).

We calculated the sample size on the basis of 80% power and a 5% two-tailed significance level. We assumed that we would need to explore modest differences of a continuous variable between two groups of unequal size. We therefore calculated the sample size on the basis of detecting a difference of half a standard deviation between two groups the smallest of which comprises 10 of the total. This yielded a required sample size of 350 (https://epitools.ausvet.com.au/twomeanstwo). We then doubled this to take into account factors such as refusals and non-response

Please include additional information regarding the survey or questionnaire used in the study and ensure that you have provided sufficient details that others could replicate the analyses. For instance, if you developed a questionnaire as part of this study and it is not under a copyright more restrictive than CC-BY, please include a copy, in both the original language and English, as Supporting Information. In addition please provide additional information regarding the interview guide development process, including the theories or frameworks which were employed.

Information added and copy of survey added 

This study is not qualitative study but a questioner base study about demographic , dietary and risk factor 

We note that you have reported significance probabilities of 0 in places. Since p=0 is not strictly possible, please correct this to a more appropriate limit, eg 'p<0.0001'.

done

Data included in the supplementary file

4. Thank you for stating the following in the Financial Disclosure section: 

We note that one or more of the authors are employed by a commercial company: Ambulatory Healthcare Services, Abu Dhabi Health services company. 

All governmental organization. no commercial company, and no fundaing 

No funding provided for conducting the study , All the author working in the government organization .

Not applicable. No specific funding for the study was received. All decisions regarding design, analysis, and reporting were solely made by the authors. None of the authors has any commercial affiliation

No commercial affiliation and no funding , all the author working in governmental organization 

Done

Reviewers' comments:

Reviewer's Responses to Questions

Comments to the Author

1. Is the manuscript technically sound, and do the data support the conclusions?

Reviewer #1: Yes

Reviewer #2: Yes

2. Has the statistical analysis been performed appropriately and rigorously?

Reviewer #1: I Don't Know

Reviewer #2: Yes

3. Have the authors made all data underlying the findings in their manuscript fully available?

Reviewer #1: Yes

Reviewer #2: Yes

4. Is the manuscript presented in an intelligible fashion and written in standard English?

Reviewer #1: Yes

Reviewer #2: Yes

5. Review Comments to the Author

Reviewer #1: The title must be changed because it is not clear

Done

Material and Methods:

I suggest to begin by stating the study design and where the study was performed. Please, specify all inclusion and exclusion criteria.

Please, specify were the study was conducted.

Done

Study design: Cross-sectional retrospective observational study conducted in Ambulatory health care services, Abu Dhabi, United Arab Emirates . 

Subject: Adults aged ≥18 years presenting to Ambulatory Health Care clinic for the Weqaya screening program between October 2011 and February 2013 were randomly recruited. The Inclusion Criteria are participants who did the Weqaya screening test and agreed to be enrolled in the study after taking verbal consent through the phone. The only exclusion criteria were refusing to participate, impossibility to reach by phone, and having chronic diarrhea, chronic liver disease, and chronic kidney disease, i.e. conditions that may severely affect Vitamin D levels

Please, specify the instrumentation for immunoturbidimetry assay.

Please, verify that the following sentence is correct “Cardiac C-Reactive Protein (Latex) High Sensitive kit (CRPHS)”.

In the material and methods, Authors should state that there was also a control cohort (non-HTN) specifying the characteristic of such population, including the criteria of exclusion and inclusion

Not in our paper ?? kindly clarify 

The discussion should be better organized and also more appropriate terms should be used.

Done modification in Discussion done 

Reviewer #2: The manuscript cover a large part of Vitamin D deficiency issus, addressing many questions in an elegant fashion and a clear presentation. However, some minor changes could improve it: please, revise the reference list according to the major works thatare available in the literature, mainly consider citing: -PMID: 18516267 -PMID: 31330127 -PMID: 28801380 -PMID: 30588177 -PMID: 31284484 -PMID: 30793054 -PMID: 31142227

Thank you for the reference , I citaed the article related to our objective and Discussion modified 

6. PLOS authors have the option to publish the peer review history of their article (what does this mean?). If published, this will include your full peer review and any attached files.

Do you want your identity to be public for this peer review? For information about this choice, including consent withdrawal, please see our Privacy Policy.

Reviewer #1: No

Reviewer #2: No

---

## [Decision Letter · Decision Letter 1]

3 Feb 2022

Risk factors for Vitamin D deficiency in Abu Dhabi Emirati population

PONE-D-21-12504R1

Dear Dr. Al Zarooni,

We’re pleased to inform you that your manuscript has been judged scientifically suitable for publication and will be formally accepted for publication once it meets all outstanding technical requirements.

Kind regards,

Carla Pegoraro

Division Editor

PLOS ONE

Additional Editor Comments (optional):

Reviewers' comments:

Reviewer's Responses to Questions

**Comments to the Author**

1. If the authors have adequately addressed your comments raised in a previous round of review and you feel that this manuscript is now acceptable for publication, you may indicate that here to bypass the “Comments to the Author” section, enter your conflict of interest statement in the “Confidential to Editor” section, and submit your "Accept" recommendation.

Reviewer #1: All comments have been addressed

Reviewer #2: All comments have been addressed

2. Is the manuscript technically sound, and do the data support the conclusions?

Reviewer #1: (No Response)

Reviewer #2: Yes

3. Has the statistical analysis been performed appropriately and rigorously? 

Reviewer #1: Yes

Reviewer #2: Yes

4. Have the authors made all data underlying the findings in their manuscript fully available?

Reviewer #1: Yes

Reviewer #2: Yes

5. Is the manuscript presented in an intelligible fashion and written in standard English?

Reviewer #1: (No Response)

Reviewer #2: Yes

6. Review Comments to the Author

Reviewer #1: The Authors revised the manuscript according to the suggestions. The manuscript is now suitable for publication.

Reviewer #2: (No Response)

7. PLOS authors have the option to publish the peer review history of their article (what does this mean?). If published, this will include your full peer review and any attached files.

Reviewer #1: No

Reviewer #2: No

---

## [Editor Report · Acceptance letter]

7 Feb 2022

PONE-D-21-12504R1 

Risk factors for Vitamin D deficiency in Abu Dhabi Emirati population 

Dear Dr. Al Zarooni:

I'm pleased to inform you that your manuscript has been deemed suitable for publication in PLOS ONE. Congratulations! Your manuscript is now with our production department. 

Kind regards, 

on behalf of

Dr Carla Pegoraro 

Staff Editor

PLOS ONE